



# Evolution of Wind Field in the Atmospheric Boundary Layer with using of Multiple Sources Observations during the Transit of Super Typhoon Doksuri (2305)

Xiaoye Wang[1,2], Jing Xu[1,2*], Songhua Wu[3,4,5*], Qichao Wang[6], Guangyao Dai[3,4], Peizhi Zhu[3], Zhizhong Su[7], Sai Chen[8], Xiaomeng Shi[2] and Mengqi Fan[6]

[1]Qingdao Joint Institute for Marine Meteorology, Chinese Academy of Meteorological Sciences, Qingdao, 266061, China
[2]Qingdao Meteorological Observatory, Qingdao Meteorological Bureau, Qingdao, 266003, China
[3]College of Marine Technology, Faculty of Information Science and Engineering, Ocean University of China, Qingdao, 266100, China
[4]Laoshan Laboratory, Qingdao, 266237, China
[5]Institute for Advanced Ocean Study, Ocean University of China, Qingdao, 266100, China
[6]Qingdao Leice Transient Technology Co., Ltd., Qingdao, 266100, China
[7]Xiamen Key Laboratory of Straits Meteorology, Xiamen Meteorological Bureau, Xiamen, 361012, China
[8]Xiamen Meteorological Bureau, Xiamen, 361012, China

*Correspondence to*: Jing Xu (xujing@cma.gov.cn) and Songhua Wu (wush@ouc.edu.cn)

**Abstract.** The accurate wind field observation of tropical cyclone (TC) boundary layer is of great significance to improve the TC track and intensity forecasting. To investigate the vertical structure of TC boundary layer during the landfall process of Super Typhoon Doksuri, three kinds of instruments including the coherent Doppler lidar (CDL), radar wind profiler (RWP) and automatic weather station (AWS) are deployed at two sites in Xiamen, Fujian province. A data fusion method is developed to obtain the complete wind speed profiles covering the whole Atmospheric Boundary Layer (ABL) based on the above instruments. The wind speeds in the near field blind zones of CDL observation are interpolated by combining the AWS measurements at 10 m. The CDL provides high temporal-spatial resolution wind speed profiles from tens of meters to its highest detection height. The wind speeds above the highest detection height of the CDL would be supplemented with the RWP measurements. The hourly mean wind speed profiles are compared with traditional models. Generally, the wind speed profiles fit well with the power law in the lower part of the ABL, before wind speed changes rapidly. However, it would cause a large error (up to 73%) to describe the exact wind speed profiles with traditional models during and after the typhoon's passage, especially when the wind speed is almost constant with height or when wind shear exists. Then fine structures and evolutionary processes of the wind field in the ABL during the typhoon landfall are investigated. In addition, the wind field distribution and wind speed variation with distance from the typhoon center are statistical analyzed. The joint wind field measurements of CDL, RWP and AWS have the broad application prospects on the dynamics study of the TC boundary layer and the improvement of the boundary layer parameterization scheme in numerical forecast models.



# 1 Introduction

As one of the most severe meteorological disasters, the tropical cyclone (TC) would cause the heavy casualties and significant economic losses in the southeastern coastal areas of China. Therefore, the accurate prediction of the TC track and intensity is vitally important. The wind field observation of the TC boundary layer and the study of its vertical structure are of great importance to improve the boundary layer parameterization scheme in numerical forecast models and to deepen the understanding of the TC evolution mechanism.

Over the past decades, researches on TC boundary layer observations have been widely conducted by radiosonde balloons, GPS dropsondes, anemometers, radar wind profilers (RWP) and weather radar. Many studies focus on the characteristics of the wind field especially the mean wind speed profiles. The obtained mean wind speed profiles have been compared with the traditional models below 500 m to evaluate the applicability of different models during different TC events (He et al., 2020; He et al., 2023b; Luo et al., 2020; Shu et al., 2017; Song et al., 2016). Turbulent parameters below 500 m are estimated from high-frequency wind measurements to analyze the kinematic and thermodynamic boundary layer structures including momentum flux (Duan et al., 2017; He et al., 2022; Li et al., 2022; Ming et al., 2023; Ming et al., 2014; Tang et al., 2018; Zhao et al., 2022; Zhou et al., 2023), turbulence intensity (Li et al., 2022; Xia et al., 2021), turbulence integral scale (Luo et al., 2020; Wang et al., 2017; Xia et al., 2021), turbulent kinetic energy (TKE) (Duan et al., 2017; He et al., 2022; He et al., 2023a; Li et al., 2022; Ming et al., 2023; Zhao et al., 2022), vertical eddy diffusivity (Ming et al., 2023; Tang et al., 2018; Zhao et al., 2022), turbulent kinetic energy dissipation rate (TKEDR) (Fang et al., 2023; Ming et al., 2023), turbulent spatial scales (Duan et al., 2017; He et al., 2022; Lan et al., 2023; Wang et al., 2017; Xia et al., 2021), gust factor (Shu et al., 2017; Xia et al., 2021; Zhou et al., 2023), low-level jet (He et al., 2023b; Li et al., 2019) and wind shear (Shu et al., 2017). And the variation of several characteristic height scales during the TC evolution was also considered such as the mixed layer depth, the inflow layer depth and the height of the maximum wind speed (He et al., 2018; Ming et al., 2015; Ming et al., 2014; Tang et al., 2018; Zhao et al., 2022). Additionally, a few aircraft campaigns have been reported to analyze the characteristics of the wind field, system-scale circulation and turbulent structures during hurricane and typhoon events (He et al., 2023a; Lussier Iii et al., 2014; Sparks et al., 2019; Tang et al., 2021; Zhang et al., 2010; Zhang et al., 2011; Zhao et al., 2020). Generally, TC boundary layer observations and researches have made significant progresses after years of development. However, the continuity, temporal-spatial resolution and vertical detection capability of TC boundary layer observations need to be further improved.

With the development of laser remote sensing technology, Doppler wind lidar has been widely used in wind field and turbulence observations due to its advantages of high accuracy, high temporal-spatial resolution, vertical-resolved and continuous capabilities. In recent years, several studies have been conducted to investigate the TC boundary layer structure using Doppler wind lidars. Aiming at typhoons Dujuan and Soudelor that occurred in 2015, Tsai et al. (2019) estimated the mean wind speed profiles within different average periods by the ground-based coherent Doppler lidar (CDL). The wind speed profiles at different evolution stages of TC development were compared with logarithmic law and power law models





under the assumption of neutral conditions. Shi et al. (2021) analyzed the characteristics of the wind field, TKE and boundary layer height scales of typhoon Lekima (2019) based on five ground-based CDLs. Similar to the above studies, Chen et al. (2023) captured typhoons In-Fa and Chanthu in 2021 with ground-based CDL observations. Considering the volume averaging effect in CDL detection, they proposed a correct method based on a wind spectrum model to improve the retrieval accuracy of turbulent parameters. Except for the ground-based CDLs, an airborne CDL system was used and the

retrieval accuracy of the wind field was validated during Tropical Storm Erika (2015). The wind speed measured by the CDL showed good agreements with that observed by the GPS dropsonde (Zhang et al., 2018). In summary, the feasibility and potential of the CDL in TC observation have been verified by the above studies. However, the detection range of CDLs is usually significantly reduced because typhoons bring heavy rainfall and the energy of the emitted laser beam is rapidly attenuated by raindrops. The wind field observations could be realized up to a few hundred meters instead of covering the

entire atmospheric boundary layer (ABL).

In this paper, combined wind field observations with CDL, RWP and automatic weather station (AWS) are conducted to study the vertical dynamic structure of the TC boundary layer during the transit of super typhoon Doksuri. The consistency of the wind field measured by the above instruments is first validated. And the no-blind zone wind profiles covering the entire ABL are obtained and compared with traditional logarithmic law, power law and other well-known models based on

the combined measurements of CDL, RWP and AWS. Then fine structures and evolutionary processes of the wind field in the ABL during the typhoon landfall are investigated in detail. In addition, the wind field distribution and wind speed variation with distance from the typhoon center are statistical analyzed.

The paper is organized as follows. The descriptions of the super typhoon Doksuri and all involved instruments are presented in Section 2. Section 3 introduces the methods used to estimate traditional wind speed profile models and no-blind zone wind

speed profiles. The comparison of the mean wind speed profiles with several well-known models and analysis of the wind field evolution processes are provided in Section 4. Then statistical analyses of the wind field distribution and wind speed variation with distance from the typhoon center are also performed in this section. Section 5 summarizes the conclusions and describes the outlook of future studies.

## 2 Typhoon and Instruments Description

### 2.1 Super Typhoon Doksuri Description

As the fifth named storm of the 2023, typhoon Doksuri formed in the northwestern Pacific Ocean on 21 July 2023. Its best-track and intensity information are displayed in Fig. 1 (a) which are from the China Meteorological Administration (CMA) TC Best Track Dataset (Lu et al., 2021; Ying et al., 2014). Typhoon Doksuri first moved northwesterly with increasing intensity and developed into a severe typhoon at 08:00 local standard time (LST, LST = UTC + 8) on 24 July. Then it

reached to a super typhoon at 20:00 LST on 24 July. Before 11:00 LST on 28 July, it remained at severe typhoon and super typhoon levels for about 96 hours. At about 09:55 LST on 28 July, severe typhoon Doksuri made landfall in Jinjiang, Fujian





province with the maximum wind speed of 50 m/s and minimum sea level pressure of 945 hPa. After landfall, it continued to move in the northwest direction and brought heavy rainfall to Fujian province even North China and Huang-Huai areas.

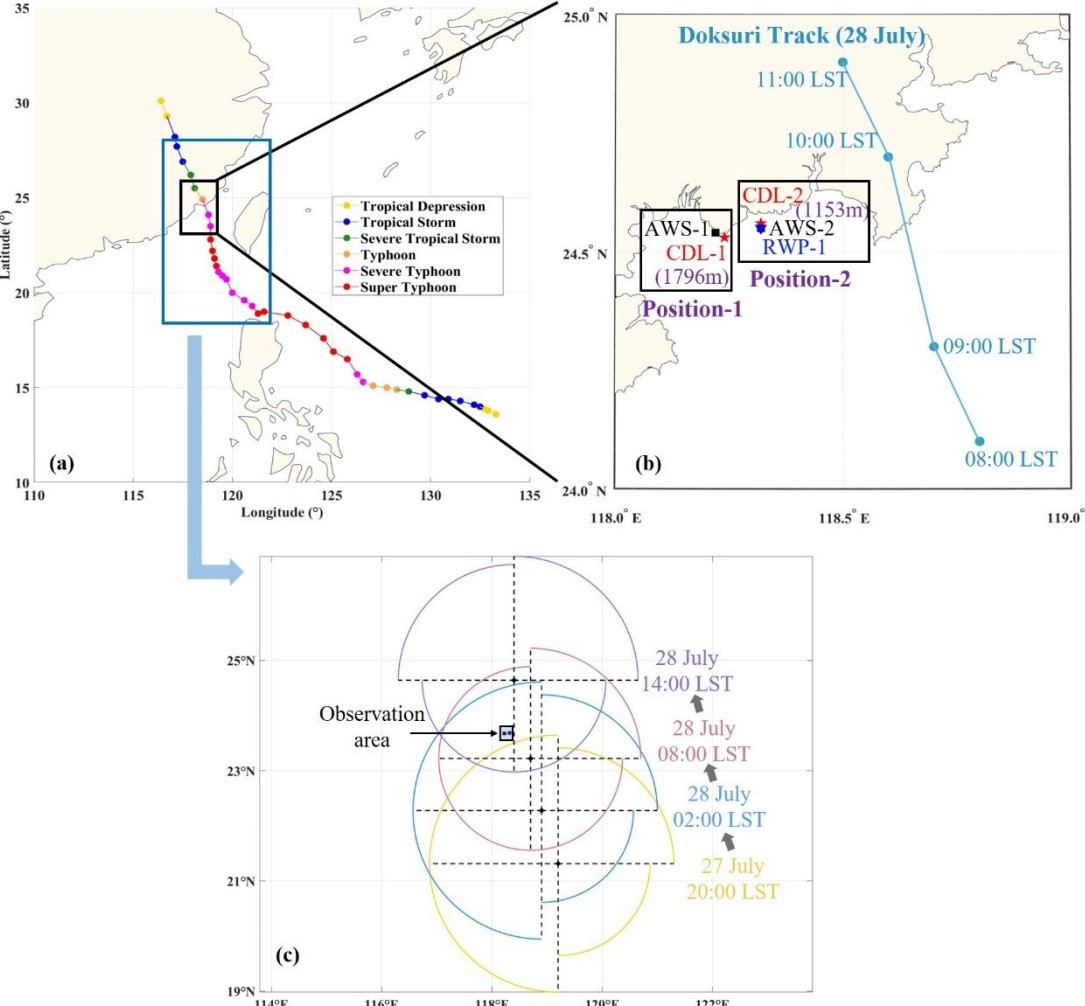

**Figure 1: (a) Best-track and intensity of Super Typhoon Doksuri, (b) the layout of the observation instruments including CDLs (red stars), AWSs (black squares), and RWP (blue triangle) and (c) the relative position of the observation area and the 34-kts wind circles.**

## 2.2 Instruments

During the transit of super typhoon Doksuri, all instruments including CDL, RWP and AWS were deployed at two sites in Xiamen, Fujian province. The layout of observation instruments is shown in Fig. 1 (b). The distances between each CDL and the nearest AWS are labeled with purple fonts. And the other details of the observation areas and key specifications of all instruments are summarized in Table 1. According to the 34-kts wind circle information from Joint Typhoon Warning Center (JTWC), the observation area was within the 34-kts wind circle from 20:00 LST on 27 July to 14:00 LST on 28 July as exhibited in Fig. 1 (c).





**Table 1: Details of the observation areas and instruments key specifications.**

| Location | | Instrument | Measurement range (m) | Temporal resolution (min) | Spatial resolution (m) |
|---|---|---|---|---|---|
| Position-1 (Coast) | 24.54°N,118.22°E | AWS-1 | 10 | 60 | / |
| | 24.53°N,118.24°E | CDL-1 | 60 ~ 4000 | 0.1 | 15 |
| Position-2 (Island) | 24.55°N,118.32°E | AWS-2 | 10 | 60 | / |
| | 24.56°N,118.32°E | CDL-2 | 75 ~ 6000 | 1 | 15 |
| | 24.55°N,118.32°E | RWP-1 | 150~10110 | 6 | 120 |

The AWS provides basic meteorological elements measurements of wind speed, wind direction, temperature, relative humidity and pressure information near the ground. Two CDLs used in this study are jointed developed by Ocean University of China (OUC) and Qingdao Leice Transient Technology Co., Ltd (http://www.leice-lidar.com/en/index.html, last access: 11 July 2024). Its principle and schematic diagram have been introduced detailly in another study (Wu et al., 2016). The type

of CDL-1 is WindMast PBL while the type of CDL-2 is Wind3D 6000, and their key specifications could be found in a separate paper (Wu et al., 2022). The RWP used belongs to the ground-based RWP network and is produced by the 23rd Institute of China Aerospace Science and Industry Corporation (http://www.casic23.com.cn/index.html, last access: 24 July 2024). Its principle and main parameters have been reported systematically in several studies (Liu et al., 2019; Guo et al., 2023).

Three-dimensional wind field profiles could be obtained by the CDL and RWP measurements with different temporal-spatial resolutions. To ensure the measurement accuracy, the quality control needs to be considered. As reported by previous studies (Jin et al., 2022; Wang et al., 2023), the CDL-measured wind field data with the SNR less than 8 dB would be eliminated to ensure the reliability. For the RWP, the horizontal speed and direction results would not be used if the horizontal credibility is less than 100%. It needs to emphasize that the measured vertical velocity is inaccurate during precipitation cause that the

Doppler spectrum of the aerosol and precipitation are detected simultaneously and have a bimodal or multimodal distribution (Aoki et al., 2016). The Doppler frequency of the precipitation and vertical velocity could be obtained accurately by fitting the two-component or multivariate Gaussian model. This method is being developed by other colleagues in our group thus the vertical velocity variation would not be analyzed in this paper.

The accuracy of the horizontal speed and horizontal direction observed by CDLs have already been evaluated with mast-

mounted cup anemometers and wind vanes at Haiyang, Shandong Province on July 2021 (Wu et al., 2022). Well agreement between them with the speed and direction correlation of 0.998, the speed standard deviation (SD) and bias of 0.14 m/s and 0.07 m/s, and the direction SD and bias of 2.75° and -1.21° is obtained. The performance of the RWP network in China has been evaluated including the system performance index and data accuracy during November 2018 to March 2019 (Liu et al., 2020). The horizontal speed results observed by RWP used in our study show good agreement with that obtained from ERA5



numerical model from 0 to 3 km. The mean speed difference (MSD) and the root-mean-square difference (RMSD) are less than 2 m/s and 4 m/s, respectively.

## 3 Methodology

### 3.1 Wind Speed Profile Models

As reported by other studies, the atmosphere could be considered under neutral or near-neutral conditions when the mean
horizontal speed exceeds a certain threshold of 6 m/s (Luo et al., 2020) or 10 m/s (Tse et al., 2013). High winds caused by the TC circulation weaken the thermal stratification, and turbulent mixing processes are dominated by wind shear (Tsai et al., 2019; Tse et al., 2013).

The Surface layer (SL) is the lower layer of the ABL, generally tens of meters in height, and varies with the height of the ABL. Assuming that the friction velocity is constant, the horizontal speed profiles satisfy the logarithmic formulation:

$$U(z) = \frac{u_*}{k} \ln\left(\frac{z}{z_0}\right),\tag{1}$$

where $U(z)$ is the horizontal speed at a certain height $z$, $u_*$ is the surface friction velocity, $k$ is the von Karman constant of 0.4 and $z_0$ is the aerodynamic roughness length. $u_*$ and $z_0$ are estimated with linear fitting in logarithmic coordinates at three lowest heights (Tsai et al., 2019):

$$U(z) = \frac{u_*}{k} \ln(z) - \frac{u_*}{k} \ln(z_0).\tag{2}$$

Above the SL, it is not appropriate to describe the horizontal speed profiles with a logarithmic law. Several well-known wind speed models are selected for comparison with the observation including the Blackadar and Tennekes (BT) model (Blackadar and Tennekes, 1968), the Deaves and Harris (DH) model (Deaves, 1978; Harris, 1981) and the Gryning model (Gryning et al., 2007). Their expressions are provided in Eq. (3) to Eq. (5):

$$U(z) = \frac{u_*}{k}\left[\ln\left(\frac{z}{z_0}\right) + \frac{kz}{\eta} - \frac{z}{z_h}\left(\frac{kz}{2\eta} + 1\right)\right],\tag{3}$$

$$U(z) = \frac{u_*}{k}\left[\ln\left(\frac{z}{z_0}\right) + \frac{z}{L_{ml}} - \frac{z}{z_h}\left(\frac{z}{2L_{ml}}\right)\right],\tag{4}$$

$$U(z) = \frac{u_*}{k}\left[\ln\left(\frac{z}{z_0}\right) + 5.75\frac{z}{z_h} - 1.88\left(\frac{z}{z_h}\right)^2 - 1.33\left(\frac{z}{z_h}\right)^3 + 0.25\left(\frac{z}{z_h}\right)^4\right].\tag{5}$$

where $\eta = k_b u_*/f_c$, $z_h = u_*/6f_c$ and $k_b = 6.3 \times 10^{-3}$. $f_c$ is the Coriolis parameter which could be calculated according to the latitude. $L_{ml}$ is the mixing length determined as Gryning et al. (2007) introduced:

$$L_{ml} = \frac{u_*/f_c}{-2\ln(u_*/f_c z_0)+55}.\tag{6}$$

Additionally, the power law model of horizontal speed has become more widely used in recent years:

$$U(z) = U_{ref}\left(\frac{z}{z_{ref}}\right)^\alpha.\tag{7}$$





In this study, we choose the lowest height that CDL detected as the reference height $z_{ref}$, and $U_{ref}$ is the horizontal speed at $z_{ref}$. Eq. (7) could be converted as following:

$$\ln\left(\frac{U(z)}{U_{ref}}\right) = \alpha \ln\left(\frac{z}{z_{ref}}\right). \tag{8}$$

Then the power exponent $\alpha$ would be obtained using linear fitting and least-squares method.

## 3.2 No-Blind Zone Wind Speed Profiles

Near field blind zones exist in the CDL detection and the wind field observation range is usually greatly reduced when a typhoon passes and brings heavy rainfall. To overcome the above problems, a data fusion method is developed to obtain the complete wind speed profiles covering the entire ABL based on the combined measurements of CDL, RWP and AWS. The
flowchart of this method is displayed in Fig. 2.

This method includes two modules for low-level and high-level. The wind speed is first averaged over one hour to remove the microscale fluctuations as previous study reported (Tsai et al., 2019). In the low-level module, the AWS and CDL data are input to obtain low-level wind speed profiles. The lowest detection height $H_{low}$ (tens of meters in most cases) could be determined from the CDL-measured mean wind speed profiles. Combining the wind speed provided by the AWS
measurements, the low-level wind speed in the range of 10 m to $H_{low}$ would be estimated with cubic spline interpolation method. For the high-level module, the highest detection height $H_{high}$ of the CDL measurements must first be confirmed. If $H_{high}$ is greater than 2 km, the mean wind speed profiles from $H_{low}$ to 2 km estimated from the CDL are output directly as the high-level wind speed profiles. On the contrary, wind speeds above the $H_{high}$ need to be supplemented with the RWP measurements. The closest height $H_{RWP}$ is required to be determined and then the high-level wind speed profiles would be
outputted by combining the wind speeds observed at heights $H_{low}$ to $H_{high}$ by CDL with those measured at heights $H_{RWP}$ to 2 km by RWP. In the end, the no-blind zone wind speed profiles are obtained by combining the low-level and high-level wind speed profiles.





Figure 2: The flowchart of the no-blind zone wind speed profile retrieval.

**4 Results and Discussion**

In this section, the near-ground meteorological conditions are given by AWS measurements firstly. The consistency of the wind field measured by the CDL, RWP and AWS is validated. And the measured wind speed profiles are compared with the traditional models during the time periods that wind speeds change rapidly. Then the fine structures and evolutionary





processes of the wind field in the ABL are analyzed during the transit of typhoon Doksuri. In the end, the statistical analyses of wind field distribution and wind speed variation with distance from typhoon center are conducted.

## 4.1 Near-ground Meteorological Conditions

The variation of meteorological elements at two sites during the passage of the typhoon Doksuri are displayed in Fig. 3. We could find that meteorological elements showed similar variation tendencies at two sites generally. The temperature started to decrease at 04:00 LST on 28 July and remained the minimum of 25°C between 08:00 LST and 10:00 LST, approximately.

Then it rose gradually and stabilized after 15:00 LST on 28 July. The RH increased rapidly during 04:00 LST ~ 06:00 LST on 28 July and then mainly stayed above 85%. As the typhoon landing, the pressured dropped to the minimum of about 980 hPa. Figure 3 (d) and Figure 3 (e) present the 10-min mean wind field results hourly. The horizontal speed variations showed M-shaped bimodal distribution. The maximum wind speeds of 18 and 50m/s occurred at 11:00 LST. For the horizontal direction, it changed by more than 100° from 08:00 LST ~ 12:00 LST on 28 July with the typhoon moving.

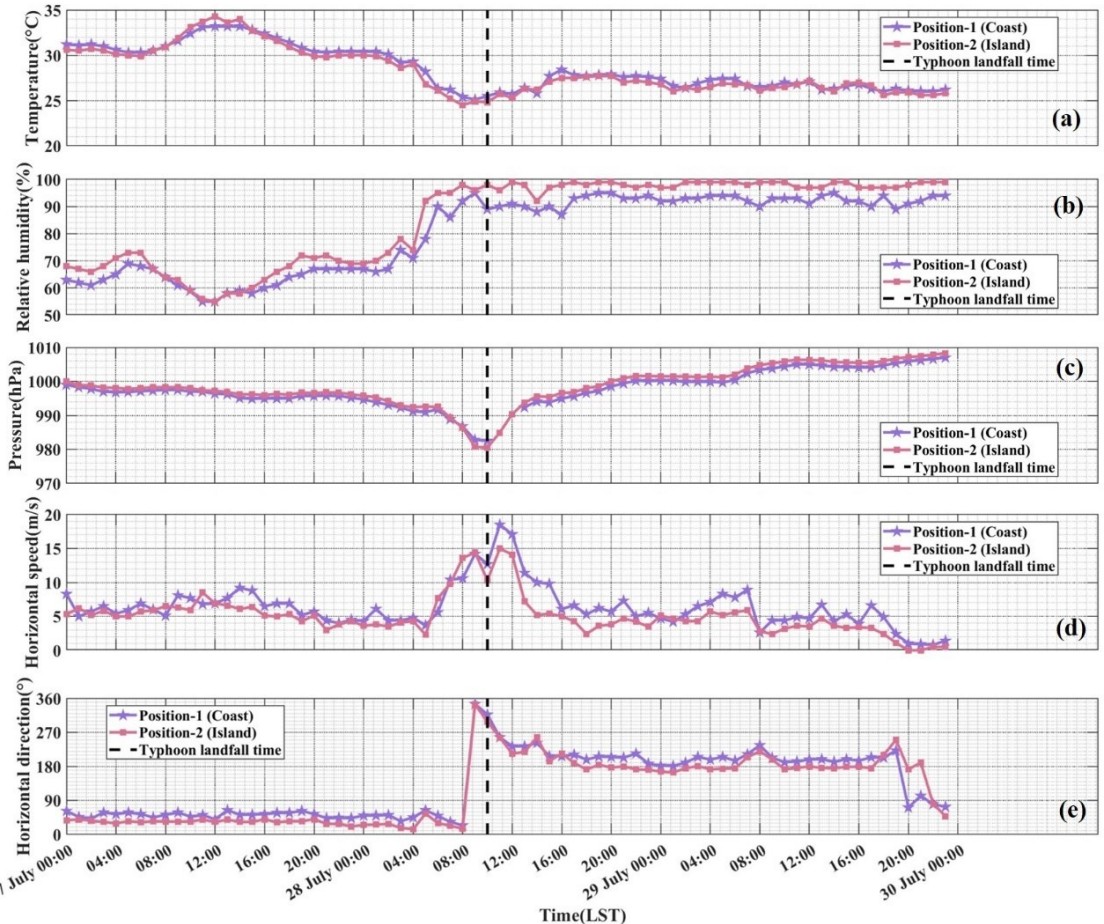


**Figure 3:** **The variation of (a) air temperature, (b) relative humidity, (c) air pressure, 10-min mean (d) wind speed and (e) wind direction measured by AWS at two locations.**



**4.2 Consistency Analysis of the Wind Field**

Before retrieving the no-blind zone wind speed profiles, the consistency of the wind speed and wind direction measured by three instruments during the passage of the typhoon Doksuri must be validated. The comparison results of the CDL and AWS measurements are exhibited in Fig. 4. The lowest detection height of the CDL-1 was 54 m while that of the CDL-2 was 71 m. The wind field results of the CDL lowest detection height and 10 m at two positions are presented in Fig. 4 (a) ~ Fig. 4 (b) and Fig. 4 (c) ~ Fig. 4 (d). In general, the variation of wind speed and wind direction at the CDL lowest detection height and 10 m showed a similar tendency and only the numerical differences existed. Hence the wind field results from 10 m to the CDL lowest detection height could be obtained with the interpolation method as Fig. 2 introduced.

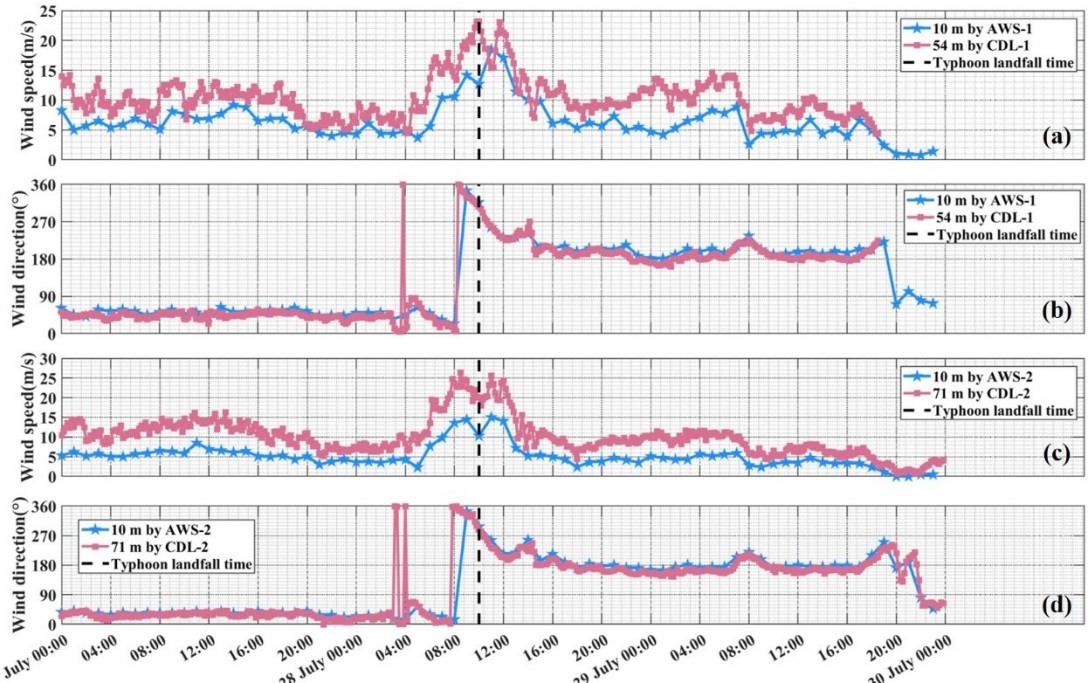

**Figure 4: Wind speed (a, c) and wind direction (b, d) variation of the CDL lowest detection height (pink squares) and 10 m (blue stars) at Position-1 (a, b) and Position-2 (c, d).**

At Position-2, a RWP (RWP-1) was deployed close to the CDL-2. Although the wind accuracies of the two instruments have been validated in detail before the typhoon observation, as described in section 2.2, their performances during the landfall of the typhoon are also compared and evaluated as shown in Fig. 5 and Fig. 6. From Fig. 5 (a) and Fig. 5 (b), the wind field profiles showed good agreement below 1500 m generally before the typhoon making landfall. Due to the high precipitation, the detection ranges of the CDL decreased to about 500 m when the typhoon landing. The horizontal speeds estimated by the CDL were consist with that measured by the RWP as displayed in Fig. 5 (c). However, from Fig. 5 (d), we could find that there were differences in the horizontal direction measurements of two instruments. The differences may result from the decrease in the horizontal homogeneity range caused by the typhoon. And the statistical results of all wind field results from





27 July to 29 July 2024 are provided in Fig. 6. Generally, the wind speed and wind direction results of the CDL and RWP measurements agreed well. The coefficient of determination and the root-mean-square error (RMSE) for the horizontal speed are 0.89 and 2.46 m/s. For the horizontal direction, the coefficient of determination and the RMSE are 0.99 and 13.35°.

Consequently, the combination of the CDL and RWP has the advantage of increasing detection ranges which would help to capture the entire vertical structure of the wind field in the typhoon observations.

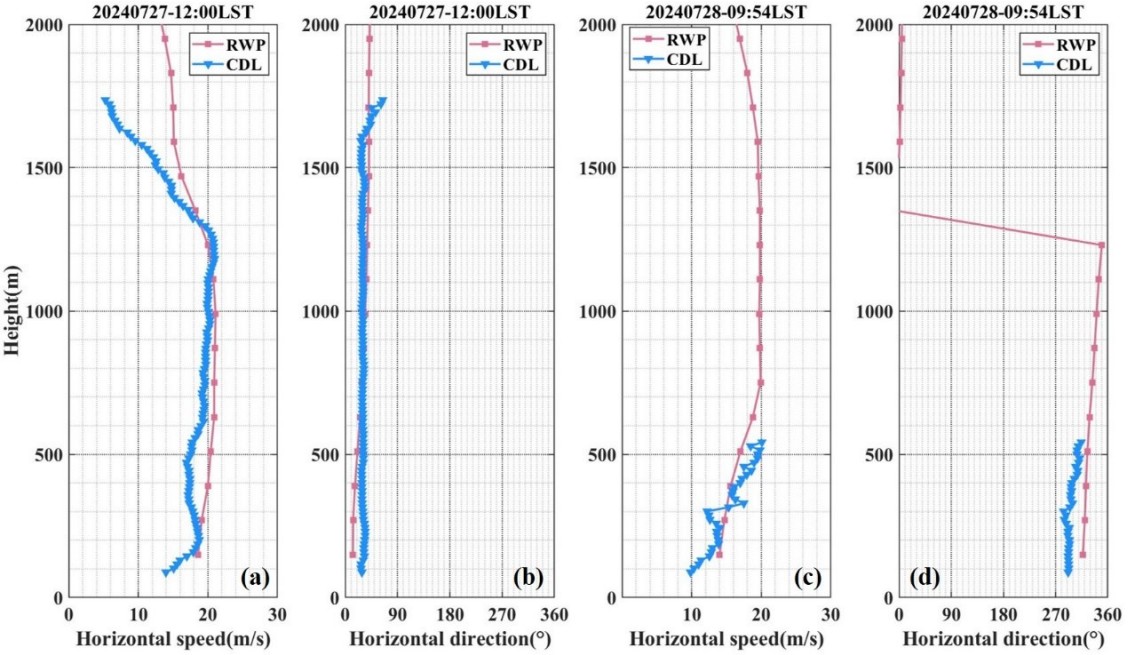

**Figure 5: Wind profile results of the horizontal speed (a, c) and horizontal direction (b, d) obtained from the CDL (blue triangles) and the RWP (pink squares) before (a, b) and during (c, d) typhoon making landfall.**

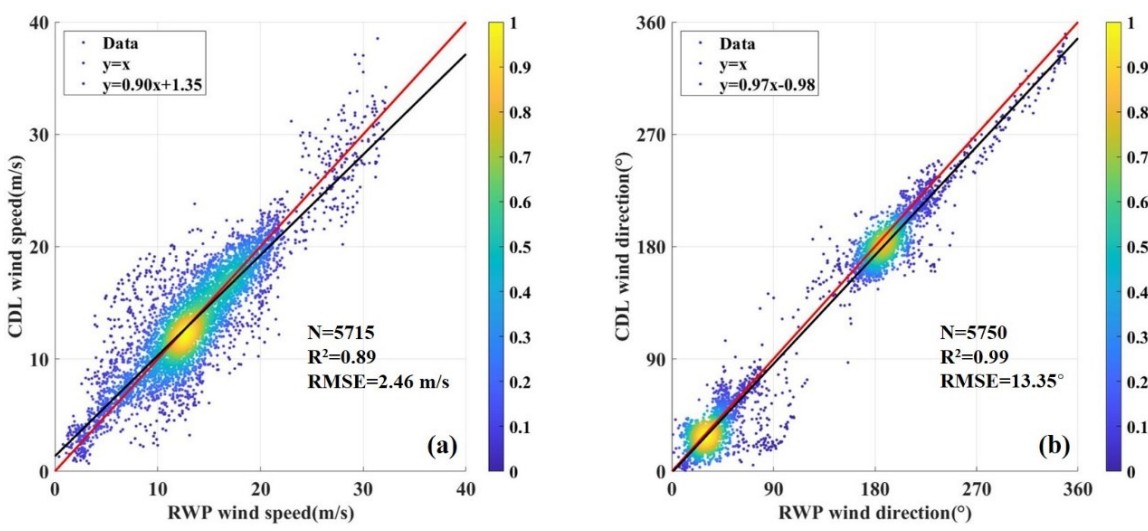


**Figure 6: Statistical results of the wind speed (a) and wind direction (b) from 27 July to 29 July 2024.**





### 4.3 Comparison of Measured Wind Speed Profiles with Models

The hourly mean wind speed profiles measured at two positions are provided in Fig. 7 and Fig. 8. Figure 7 exhibits the wind speed profiles observed by CDL-1 during 05:00 LST ~ 14:00 LST on 28 July as the wind speed changed rapidly in this

period. Before 07:00 LST, the thickness of SL was about 200 m because the wind speed profiles below 200 m fitted well with logarithmic law. And the wind speed showed best agreement with the power law generally although it also agreed well with the BT model from 06:00 LST to 07:00 LST. During the period of 07:00 LST ~ 09:00 LST, the wind speed increased sharply with height thus almost all existing models could not describe it well. When the typhoon landing, the differences of wind speed at each altitude decreased obviously and the wind speed even became almost constant with height during 10:00

LST ~ 11:00 LST. After the typhoon making landfall, wind shear layers appeared at 600 m during 11:00 LST ~ 13:00 LST and 1100 m during 12:00 LST ~ 13:00 LST.

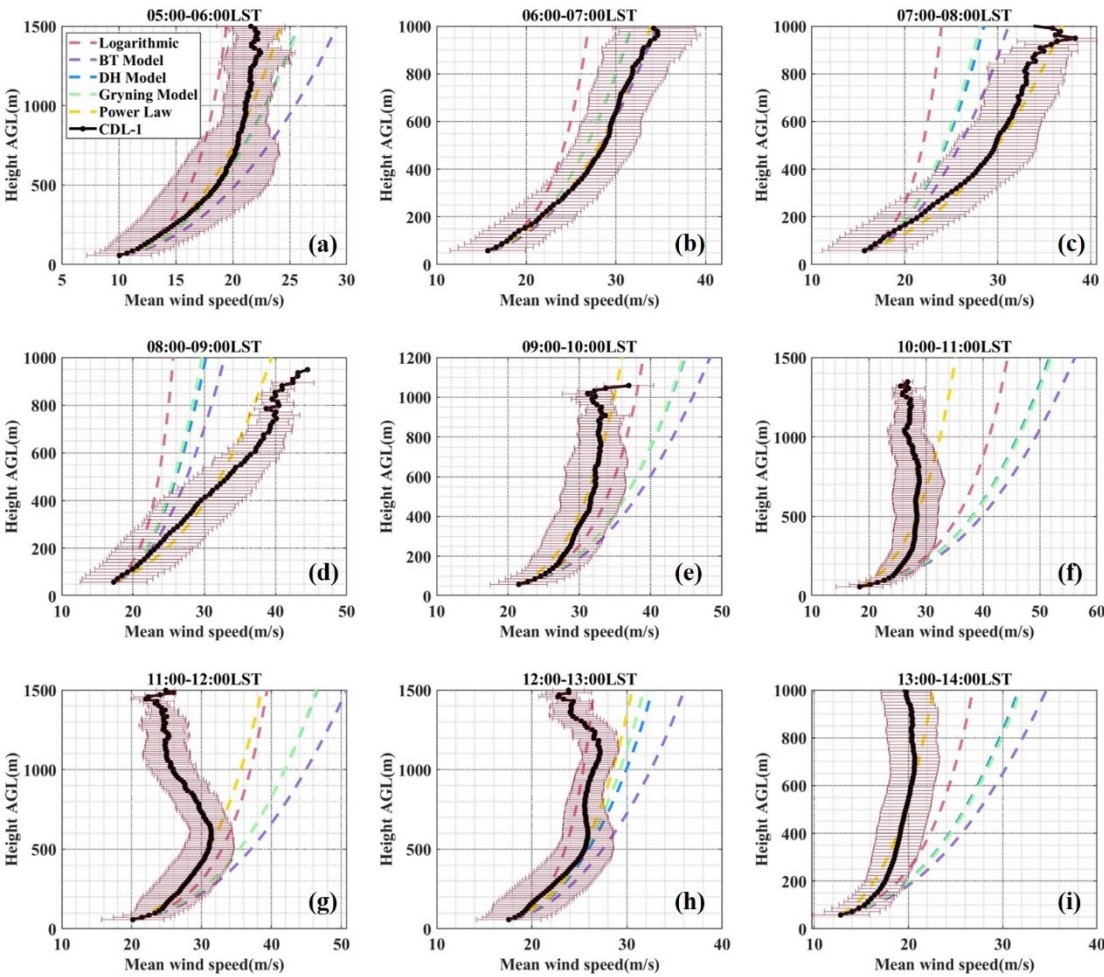

**Figure 7: Mean wind speed profiles measured by CDL-1 (black dots) and fitted by the logarithmic law (pink dotted line), BT model (purple dotted line), DH model (blue dotted line), Gryning model (green dotted line) and power law (yellow dotted line). The**
**error bars represent the standard deviation (SD) of the wind speed in the mean time period shown above.**





The no-blind zone wind speed profiles retrieved with the data fusion method are displayed in Fig. 8. The wind speed profiles agreed well with logarithmic law below 200 m before 07:00 LST. The rapid increase in wind speed kept from 06:00 LST to 10:00 LST which showed longer duration than that appeared at Position-1. During the periods of 08:00 LST to 09:00 LST, the decrease of wind speed caused by wind shear occurred in the range of 260 m to 320 m, approximately. And then wind
shear layers occurred at several heights until 14:00 LST. Overall, the variation of wind speed profiles exhibited similar characteristics with that observed by CDL-1 after typhoon landing.

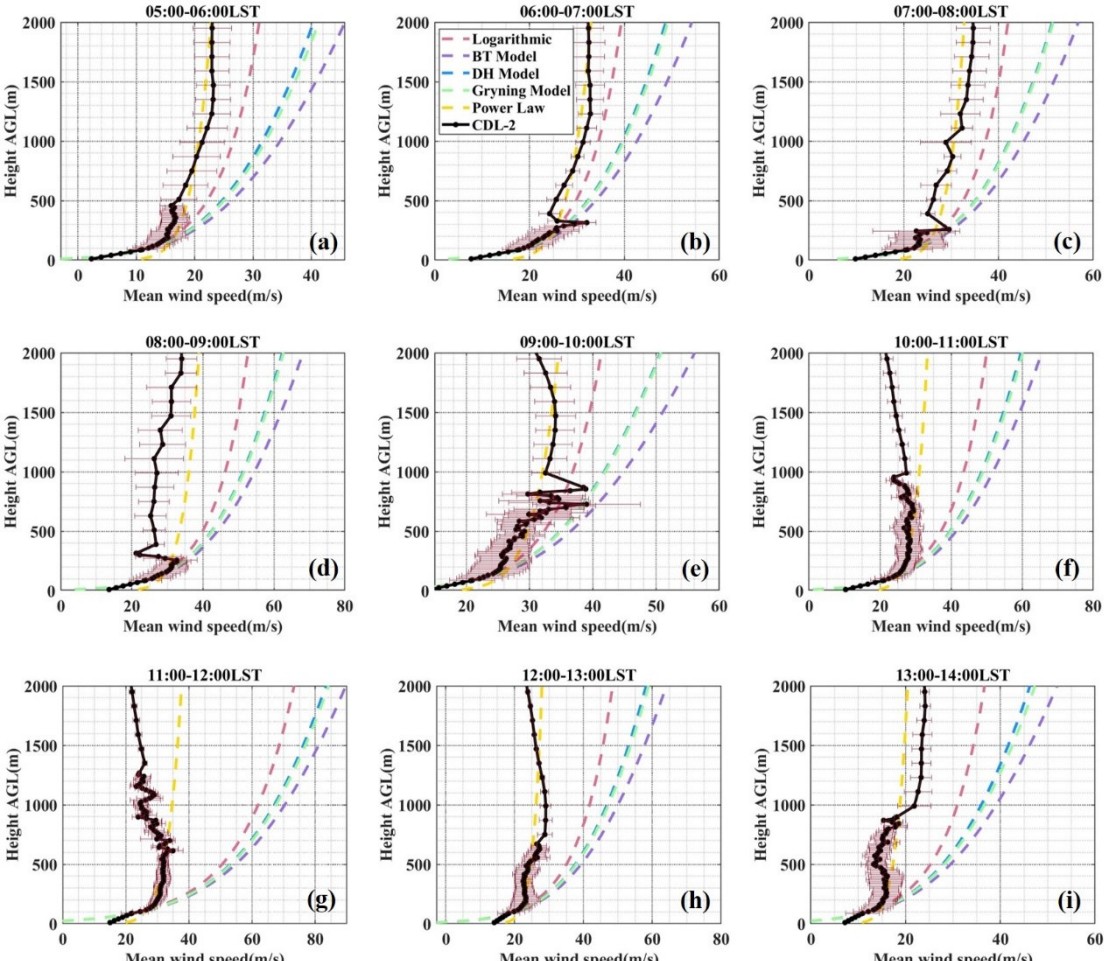

**Figure 8: No-blind zone wind speed profiles at Postion-2 using data fusion method (black dots) and fitted by the logarithmic law (pink dotted line), BT model (purple dotted line), DH model (blue dotted line), Gryning model (green dotted line) and power law**
**(yellow dotted line). The error bars represent the standard deviation (SD) of the wind speed in the mean time period shown above.**

In general, it would cause a large error (up to 73%) to describe the exact wind speed profiles with traditional models during and after the typhoon's passage, especially when the wind speed is almost constant with height or when wind shear exists. Hence the wind field measurements are still pretty important for improving real-time intensity forecasts and understanding of





the wind filed structure when typhoon passing. In future studies, we will also try to improve the traditional models based on
the measured wind profiles to achieve a more accurate description of the typhoon boundary layer structure.

## 4.4 Evolution of Wind Field during Typhoon Transit

Figure 9 provides the variation of wind speed and wind direction measured by CDL-1 from 27 July to 29 July 2024. On 27 July, the high wind speed in the ABL varied in the range of 15 m/s ~ 25 m/s and northeasterly winds dominated within the ABL. The maximum speed appeared at about 1000 m approximately. After about 06:00 LST on 28 July, the wind speed first 265    increased sharply in the upper part of the ABL and then high speed extended downwards to the SL. At 08:15 LST, the observed wind speed reached its maximum of 51.25 m/s at 442 m and then decreased after the typhoon landing. During the landfall process of the typhoon, the prevailing winds changed from northeasterly to northwesterly and then stabilized at southwesterly.

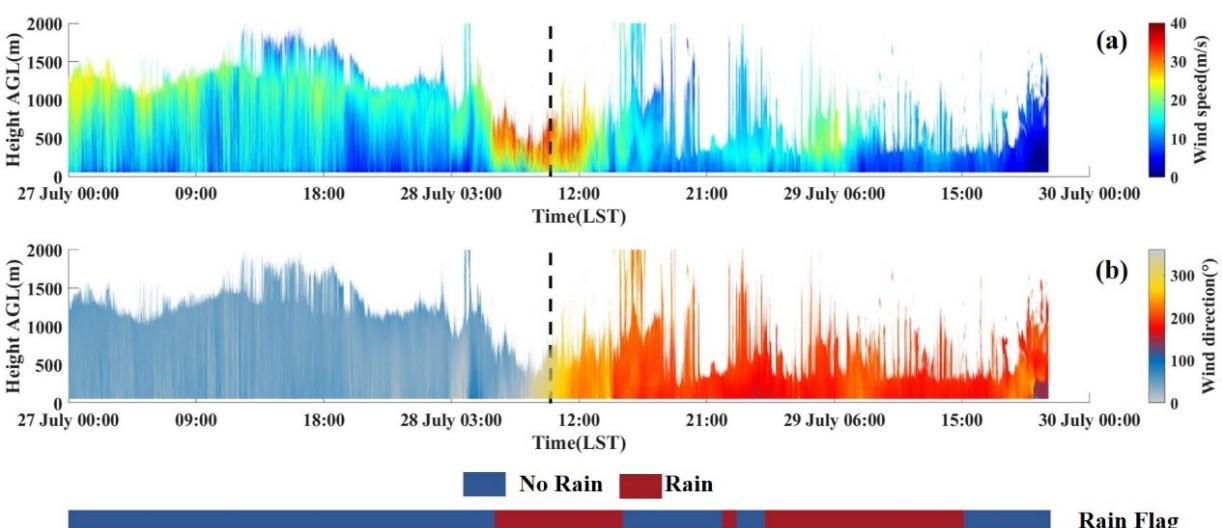

**Figure 9: The variation of (a) wind speed and (b) wind direction measured by CDL-1 during the passage of the Typhoon.**

The evolution of wind field measured by CDL-2 is exhibited in Fig. 10. Affected by the high precipitation, the detection range decreased sharply during 05:00 LST ~ 15:00 LST on 28 July. Combining the wind field results obtained by the RWP-1, the complete variation of the wind speed and wind direction covering the entire ABL is shown in Fig. 11. The variation of wind speed and wind direction was similar to that at Position-1 generally. During 07:00 LST ~ 10:00 LST, high winds within 275    the ABL exceeded 30 m/s. At 08:28 LST on 28 July, the maximum wind speed of 52.34 m/s was recorded at 571 m.





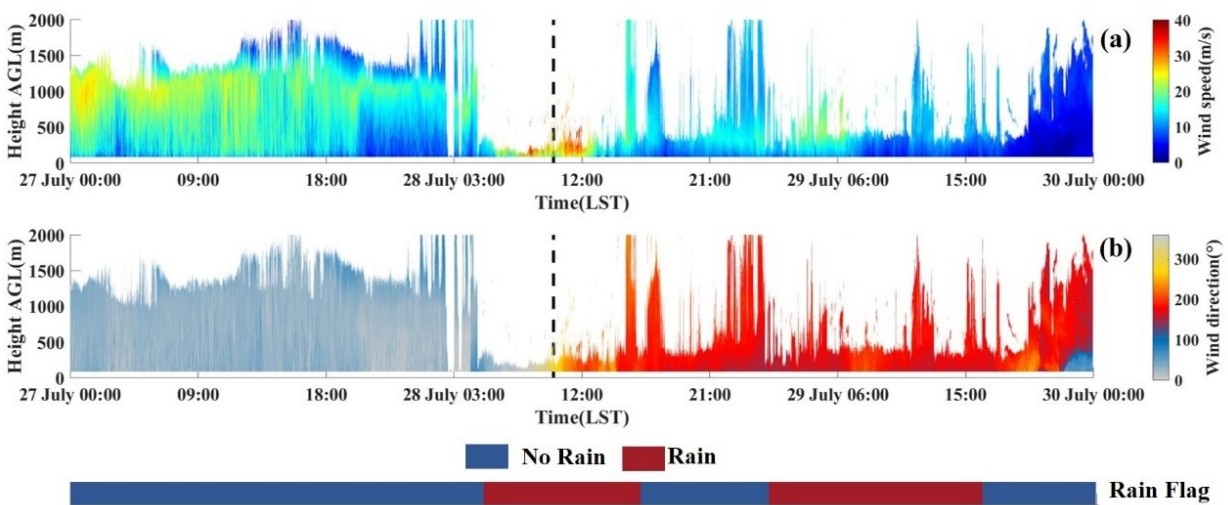

**Figure 10: The variation of (a) wind speed and (b) wind direction measured by CDL-2 during the passage of the Typhoon.**

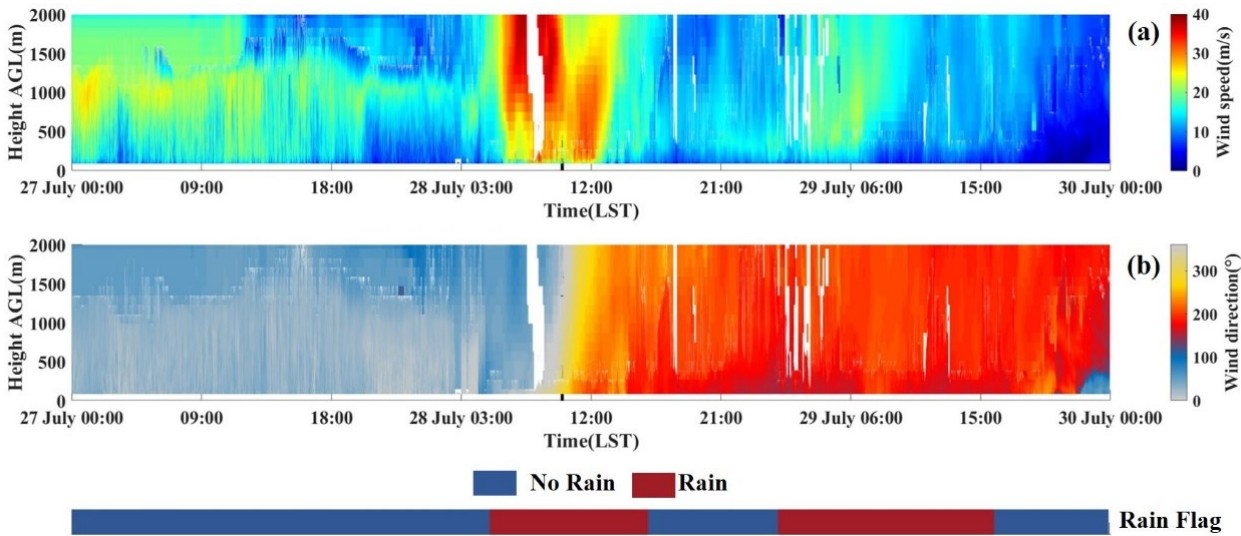

**Figure 11: The variation of (a) wind speed and (b) wind direction observed by CDL-2 and RWP-1 during the passage of the**
**Typhoon.**

## 4.5 Statistical Analysis of wind field

In order to analyze the wind field distribution characteristics, the wind speed and direction results at both positions are collected from 27 July to 29 July. Figure 12 (a) and Figure 12 (b) present the wind rose map at Position-1 and Position-2, respectively. Influenced by the track of typhoon Doksuri and the observation areas, northeasterly, southwesterly and westerly

winds dominated at Position-1 while northeasterly, westerly and easterly winds prevailed at Position-2. The wind speed was mostly concentrated in the range of 10 m/s to 20 m/s at both sites. However, the wind speed less than 10 m/s and ranging





from 20 m/s to 30 m/s covered a large proportion at Position-2, indicating that the wind speed showed a more discrete distribution.

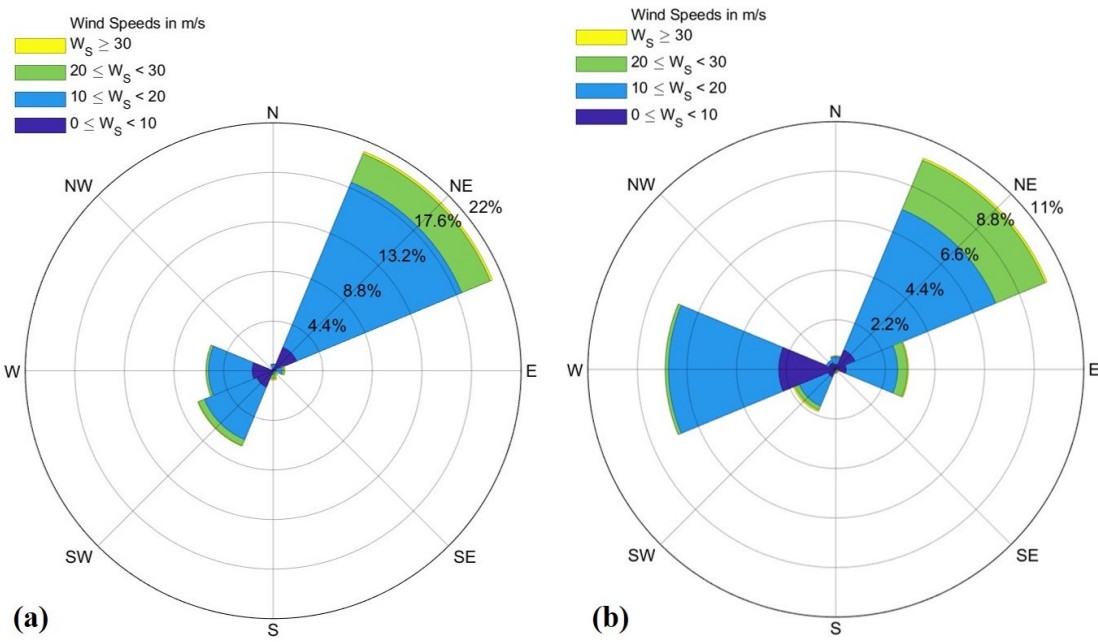

Figure 12: Wind rose map at (a) Position-1 (Coast) and (b) Position-2 (Island).

The statistical results of the observed wind speeds at different distances from the typhoon center are given in Fig. 13 (a) and Fig. 13 (b). The negative and positive distances are set to distinguish the results before and after the typhoon landing. Before the typhoon making landfall, the wind speeds increased rapidly and showed a broader distribution when the distances from the typhoon center were less than 150 km. After the typhoon passing, the wind speeds decreased sharply with the increasing of the distance from the typhoon center until the distances from the typhoon center were greater than 100 km. Then the wind speeds increased briefly and decreased slowly in general. Unfortunately, there were still lack of wind field observations of typhoon eye although the closest CDL-2 to the typhoon center was only 32 km away. In future studies, the combination of three instruments will be deployed at more appropriate locations according to the typhoon forecast to achieve the wind field measurements in the typhoon eye areas.





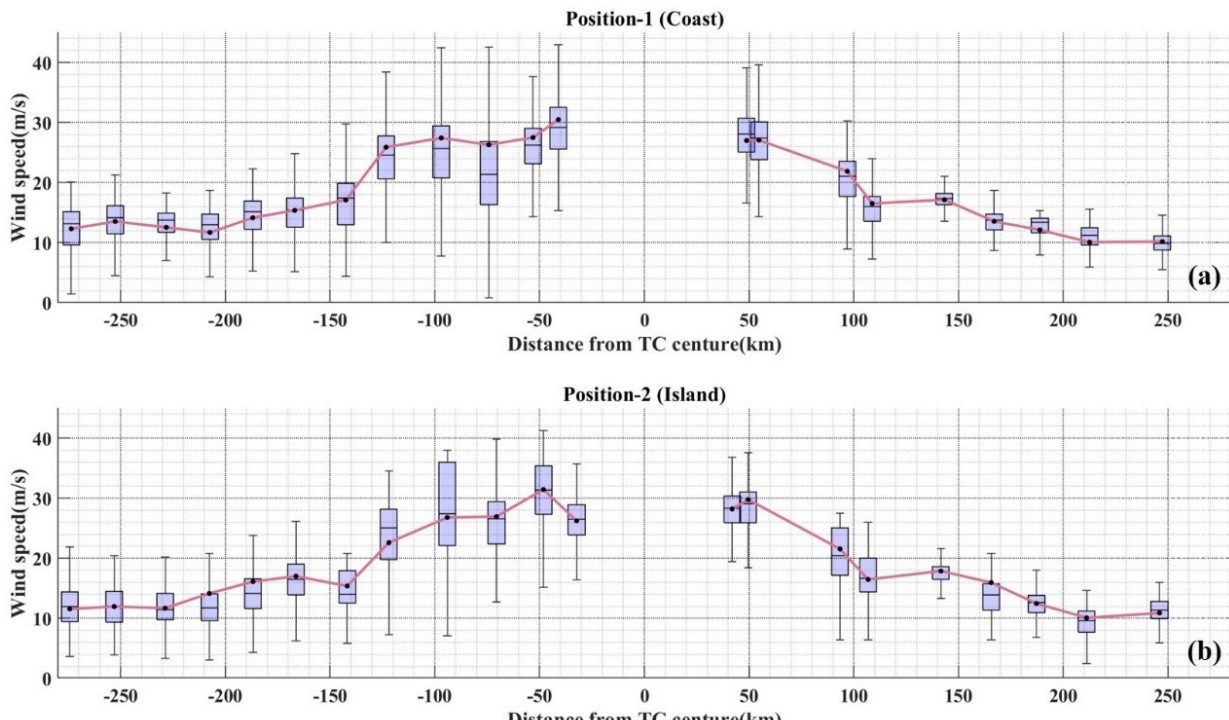

**Figure 13: The box-and-whisker plots of wind speed and its mean values (black dots) observed at (a) Position-1 and (b) Position-2 at different distances from the typhoon center.**

## 5 Conclusions

In this paper, based on the combined measurements of CDL, RWP and AWS, fine structures and evolution processes of wind field characteristics covering the entire ABL are systematically investigated during the landfall of Super Typhoon Doksuri. The key conclusions are summarized as below:

(1) The CDL system has been verified to be an effective instrument for the wind field observation of TC boundary layer. It provides high-accuracy, high temporal-spatial resolution and vertical-resolved wind field measurements which has been validated by the comparison with in-situ instruments and the nearby deployed RWP. Well agreement between the CDL and RWP is obtained during the typhoon passage with the speed and direction correlations of 0.89 and 0.99, the RMSE of 2.46 m/s and 13.35°.

(2) It is limited to perform the typhoon boundary layer observation with single instrument. Aiming at the problems of near-field blind zone and low detection range of high precipitation in CDL observation, a data fusion method is developed and applied into typhoon Doksuri observation to obtain the complete wind speed profiles covering the whole ABL based on the combined measurements of CDL, RWP and AWS. It could become a powerful complement to the existing observation methods and have the broad application prospects on the study of typhoon structure.

(3) The hourly mean wind speed profiles are compared with the traditional models. In general, the wind speed profiles show the best agreement with the power law in the lower part of the ABL before wind speed changing rapidly. However, it would cause a large error (up to 73%) to describe the exact wind speed profiles with traditional models in the upper part of the ABL,

especially during and after the typhoon passing. As the typhoon landing, the differences in wind speed at each height decrease obviously and the wind speeds even become almost constant with height. And wind shear or low-level jets occur at several heights after the typhoon landing. Therefore, the wind field measurements are still pretty important for the improvement of real-time intensity forecasts and understanding of wind filed structure.

(4) During the transit of Super Typhoon Doksuri, the wind speed and wind direction measured at two sites show similar

variation tendencies. On 27 July, the wind speed ranges from 15 m/s to 25 m/s and northeasterly winds prevail in the ABL. During 06:00 LST ~ 14:00 LST on 28 July, the wind speed first increases sharply in the upper part of the ABL and then high speed extends downwards to the SL. The measured maximum wind speed values of 51.25 m/s and 52.34 m/s appear on 08:15 LST (442 m) and 08:28 LST (571 m) at two locations, respectively. The prevailing winds change from northeasterly to northwesterly and then stabilize at southwesterly.

As an outlook, the joint wind field measurements of CDL, RWP and AWS have the broad application prospects on the dynamics study of the TC boundary layer and the improvement of the TC track and intensity forecasting. And it is promising to realize the convergence and divergence characteristic analysis within the ABL based on the multi-instrument network observations, which is of great importance to recognize the convection initiation and development of storms.

**Data Availability**

Due to confidentiality agreements, supporting data can only be made available to bona fide re-searchers subject to a non-disclosure agreement. To get the data please contact to wangxiaoye@cma.gov.cn at Qingdao Joint Institute for Marine Meteorology.

**Author contribution**

X.Wang, J. Xu, S. Wu and G. Dai conceived of the idea for the data fusion based on the joint measurements of CDL, RWP

and AWS; X.Wang wrote the manuscript; X.Wang, G. Dai, P. Zhu and X. Shi conducted the data analyses; Q. Wang, S. Chen and M. Fan collected the CDL data; Z. Su collected the RWP and AWS data; all the co-authors discussed the results and reviewed the manuscript.

**Competing interests**

The authors declare that they have no conflict of interest.




**Acknowledgments**

This study is jointly supported by the National Key Research and Development Program of China under grant 2022YFC3004200, Basic Research Fund of Chinese Academy of Meteorological Sciences under grant 2024Y015 and

Shandong Provincial Key Research and Development Program (Major Scientific and Technological Innovation Project) under grant 2023CXGC010408.

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
