# Peer review of "Evolution of Wind Field in the Atmospheric Boundary Layer with using of Multiple Sources Observations during the Transit of Super Typhoon Doksuri (2305)"

_Atmospheric Measurement Techniques, 2024_

## Referee Comment (RC1)

I have reviewed the manuscript 'Evolution of Wind Field in the Atmospheric Boundary Layer with using of Multiple Sources Observations during the Transit of Super Typhoon Doksuri' by Wang et al. The authors investigated the vertical structure of tropical cyclone (TC) boundary layer during Super Typhoon Doksuri using coherent Doppler lidar (CDL), radar wind profiler (RWP) and automatic weather station (AWS). The wind speed profiles obtained using a data fusion method fit well with traditional models in the lower part of the Atmospheric Boundary Layer (ABL) before wind speed changes rapidly. In general, the topic of this study is interesting; however, there are some unclear points that require clarification. I recommend it for publication after major revision. Please consider the following comments in revising the manuscript.

Major Comments:
Line 198: "The maximum wind speeds of 18 and 50m/s occurred at 11:00 LST". I don't see it from Figure 1d.
Figure 4: There is a sharp increase of wind direction at 54 m by CDL-1  and 71 m by CDL-2 at 04:00 LST on 28 July, which is not at 10 m by AWE-1 and AWE-2. It would be helpful that authors provide few sentences explaining that.
Lines 256-257: "In general, it would cause a large error (up to 73%) to describe the exact wind speed profiles with traditional models during and after the typhoon's passage, especially when the wind speed is almost constant with height or when wind shear exists". How did the authors get 73%? From Figure 8? Also, the sentence is unclear. Figure 8 indicates that the difference between the exact wind speed profiles and traditional models is large above 200 m, even before the typhoon's passage. Since this is one of the major conclusions, it would be helpful if the authors rephrase the sentence for clarity.

Minor Comments:
Line 196: "the pressured dropped …" -> "the pressure dropped …".

---

## Author Comment (AC1)

**Response to Anonymous Referee #1:**

**General comments:**

I have reviewed the manuscript 'Evolution of Wind Field in the Atmospheric Boundary Layer with using of Multiple Sources Observations during the Transit of Super Typhoon Doksuri' by Wang et al. The authors investigated the vertical structure of tropical cyclone (TC) boundary layer during Super Typhoon Doksuri using coherent Doppler lidar (CDL), radar wind profiler (RWP) and automatic weather station (AWS). The wind speed profiles obtained using a data fusion method fit well with traditional models in the lower part of the Atmospheric Boundary Layer (ABL) before wind speed changes rapidly. In general, the topic of this study is interesting; however, there are some unclear points that require clarification. I recommend it for publication after major revision. Please consider the following comments in revising the manuscript.

**Major comments:**

1. Line 198: "The maximum wind speeds of 18 and 50m/s occurred at 11:00 LST". I don't see it from Figure 1d.

AR: Thanks for your suggestion. This sentence has been revised to "The maximum wind speeds of 18.50m/s occurred at 11:00 LST." And it could be found in Figure 3 (d).

2. Figure 4: There is a sharp increase of wind direction at 54 m by CDL-1 and 71 m by CDL-2 at 04:00 LST on 28 July, which is not at 10 m by AWE-1 and AWE-2. It would be helpful that authors provide few sentences explaining that.

AR: Thanks for your suggestion. Actually, the wind direction measured by two CDLs showed a sharp increase because that the ordinates are set from 0° to 360°. Wind direction of 0° and 360° both refer to the north wind. Thus, there was no abrupt change in wind direction and the variation of it was less than 10°.

Additionally, the wind direction differences observed by the CDL and the AWS may be due to the differing temporal resolution. The wind directions observed by the above two instruments were averaged over 10 min and 1 h separately. Hence it is reasonable that the obvious differences in wind direction occurred especially when north wind

dominated and the wind speed varied in 0° to 10° and 350° to 360°.

Considering your comments, some explanations have been added in the section 4.2 as following "The wind direction measured by two CDLs showed a sharp increase at about 04:00 LST on 28 July because that the ordinates are set from 0° to 360°. Actually, there was no abrupt change in wind direction and the variation of it was less than 10°. When north wind dominated and the wind speed varied in 0° to 10° and 350° to 360°, it is reasonable that the wind direction measured by the CDL and the AWS exhibited obvious differences as the obtained wind directions were averaged over 10 min and 1 h, respectively."

3. Lines 256-257: "In general, it would cause a large error (up to 73%) to describe the exact wind speed profiles with traditional models during and after the typhoon's passage, especially when the wind speed is almost constant with height or when wind shear exists". How did the authors get 73%? From Figure 8? Also, the sentence is unclear. Figure 8 indicates that the difference between the exact wind speed profiles and traditional models is large above 200 m, even before the typhoon's passage. Since this is one of the major conclusions, it would be helpful if the authors rephrase the sentence for clarity.

AR: Thanks for your question and suggestion. As exhibited in Figure 7 and Figure 8, the measured wind speed profiles show the best agreement with the power law generally. Therefore, we evaluate the estimation error between the measured wind speed and the power law model. It could be found that the largest error occurred at 1435 m during the period of 11:00 LST ~ 12:00 LST at Position 1 from Figure 7 (g). The measured wind speed was 22 m/s while the wind speed estimated from the power law model was 38 m/s, and the error of 73% could be calculated as:

$$\varepsilon = \frac{38 - 22}{22} \approx 73\%$$

The conclusions have been revised both in section 4.3 and section 5 as following:

Section 4.3: "it would cause a large error (up to 73%) to describe the exact wind speed with power law model, especially when the wind speed is almost constant with height

or when wind shear exists."

Section 5: "And it would cause a large error (up to 73%) to describe the exact wind speed in the upper part of the ABL. As the typhoon landing, the differences in wind speed at each height decrease obviously and the wind speeds even become almost constant with height. And wind shear or low-level jets occur at several heights after the typhoon landing."

**Minor comments:**

1. Line 196: "the pressured dropped …" -> "the pressure dropped …".

AR: Revised. Thanks for your suggestion.

---

## Author Comment (AC2)

**Response to Anonymous Referee #2:**

**General comments:**

This manuscript presents a data fusion method to obtain the complete wind speed profiles with the combination observation of the CDL, RWP and AWS. And this method was applied and showed good performance during the landfall process of Super Typhoon Doksuri. As CDL, RWP and AWS are widely deployed in the southeastern coastal areas of China, I think this method has practical value and broad application prospects in the wind field observation within the typhoon boundary layer. The main topic of this manuscript is of interest to the readership of AMT. I recommend this manuscript for publication after addressing the following comments.

1. The core of this paper lies in the data fusion of CDL and RWP. How is the wind profile retrieved in the transition zone (or the overlapping data region) of the data? Has an assessment been conducted to evaluate the consistency of wind speed and direction observations between the two instruments?

AR: Thanks for your question and suggestion. The wind profiles in the overlapping data region are retrieved based on the CDL measurements for two reasons. As mentioned in section 2.2, the accuracies of wind field detection of the CDL and RWP have been evaluated with mast-mounted cup anemometers and ERA5 numerical model, respectively. The horizontal speed and horizontal direction measured by the CDL have higher detection accuracy. Additionally, it could be found from Table 1 that wind field would be retrieved by the CDL with higher temporal and spatial resolution of 1min and 15m.

Actually, the consistency of wind speed and direction observations between the two instruments have been evaluated detailly in section 4.2. The wind field profiles were compared both before and during the landfall of the typhoon. As shown in Figure 5, The wind field profiles generally showed good agreement below 1500 m before the typhoon making landfall while some differences existed in the horizontal direction measurements when the typhoon landing. And then all the wind filed results obtained by the CDL and RWP were compared and statistically analyzed in Figure 6. Generally, the wind speed and wind direction results of two instruments agreed well. The

coefficient of determination and the root-mean-square error (RMSE) for the horizontal speed are 0.89 and 2.46 m/s. For the horizontal direction, the coefficient of determination and the RMSE are 0.99 and 13.35°.

2. In section 4.3, you mentioned that the traditional models cause a large error (up to 73%) to describe the exact wind speed profiles during and after the typhoon's passage, especially when the wind speed is almost constant with height or when wind shear exists. How was the 73% error calculated? It would be better to give a more detailed error analysis.

AR: Thanks for your question and suggestion. As exhibited in Figure 7 and Figure 8, the measured wind speed profiles show the best agreement with the power law generally. Therefore, we evaluate the estimation error between the measured wind speed and the power law model. It could be found that the largest error occurred at 1435 m during the period of 11:00 LST ~ 12:00 LST at Position 1 from Figure 7 (g). The measured wind speed was 22 m/s while the wind speed estimated from the power law model was 38 m/s, and the error of 73% could be calculated as:

$$\varepsilon = \frac{38 - 22}{22} \approx 73\%$$

In order to evaluate the relative error caused by the power law model, we calculate the probability distribution of the relative errors at different height ranges based on the measured wind profiles in Figure 7 and Figure 8 as shown below.

[Figure]

Figure 1. The probability distribution of relative errors at (a) 0 ~ 500 m, (b) 500 ~

1000 m, (c) 1000 ~ 1500 m, (d) 1500 ~ 2000 m and (e) all heights estimated by the power law model. And the probabilistic frequencies of each group are labelled with black numbers on the histogram.

The corresponding detailed error analysis is also added as following: "Generally, the wind speed showed best agreement with the power law model. In order to evaluate the relative errors of the power law model in estimating the wind speed, the probability distribution of the relative errors at different height ranges based on the measured wind profiles in Fig. 7 and Fig. 8 are given in Fig. 9. Below 500 m, the relative errors in wind speed estimated by the power law model were small and 90% of the model results had relative errors less than 10%. The relative error of the estimated wind speed increased with height. Above 1000 m, the estimated wind speed with relative error greater than 40 % increased significantly."

3. In Line 271 to line 272, you mention that the detection range of the CDL decreased sharply affected by the high precipitation. Have you tried to utilize other techniques, such as raindrop size distribution measurements, to mitigate the impact of precipitation on the performance of CDL?

AR: Thanks for your question and suggestion. Precipitation has a great impact on CDL detection performance. On the one hand, raindrop particles would cause rapid attenuation of the laser energy thus leading to a sharp decrease in the detection range, and this effect can not be avoided. On the other hand, the falling speed of raindrops affects the detection accuracy of the vertical velocity. The Doppler frequency of the raindrops and vertical velocity could be obtained accurately by fitting the two-component or multicomponent Gaussian model as previous studies reported (Aoki et al., 2016; Liu et al., 2024).

This method has been developed by other colleagues in our group (Liu et al., 2024). Actually, a Doppler lidar power spectrum is the superposition of the signals from aerosols, raindrops, and spectral noise during rain periods. The lidar power spectrum show double-peak or multi-peak structure as shown in Figure 2. When using vertical pointing mode, the rain and vertical velocity spectra could be identified with the multicomponent gaussian model. And the nonlinear least squares algorithm of

Levenberg–Marquardt (L–M) is used to fit the Gaussian model.

[Figure]

Figure 2. Rain and wind spectra identification using the multicomponent gaussian model. The results of double-peak structure with the (a) upward and (b) downward laser beam. The multipeak structure with the (c) upward and (d) downward laser beam.

However, the amount of Doppler spectrum data is too large and it requires special settings for storage. During the passage of Typhoon Doksuri, the CDL performed automated observation and only radial velocity data was available. The correction of vertical velocity could not be realized without Doppler spectrum data. In future studies, we would try to store the Doppler spectrum data as the typhoon passing and retrieve accurate vertical velocity profiles with the method mentioned above.

Reference:

[1] Aoki, M., Iwai, H., Nakagawa, K., Ishii, S., & Mizutani, K. (2016). Measurements of rainfall velocity and raindrop size distribution using coherent Doppler lidar. Journal of Atmospheric and Oceanic Technology, 33(9), 1949-1966.

[2] Liu, X., Hasager, C. B., Mann, J., Sjöholm, M., & Wu, S. (2024). Rain measurement using nacelle-mounted Doppler lidar. IEEE Transactions on Geoscience and Remote

Sensing.

4. In this manuscript, the evolution characteristics of the vertical profiles of horizontal wind speed and direction at two stations were analyzed. Have you attempted to utilize multi-station observations to investigate the three-dimensional structure of the typhoon, such as vortex characteristics?

AR: Thanks for your question and suggestion. The boundary layer vortex characteristics containing convergence and divergence have a significant impact on the typhoon evolution. However, it is difficult to study vortex characteristics due to the lack of wind field vertical observations with high temporal and spatial resolution.

Liu et al. (2017) reported triangle method and finite element method for determining horizontal divergence and vorticity with RWP observations at three sites. The coordinate and horizontal speed of three RWP sites are expressed as $(x_i, y_i)$ and $(u_i, v_i)$, $i = 1,2,3$. When assuming a linear variation in wind speed at the three sites, the wind field at three sites would be calculated as:

$$\begin{cases}(x_2 - x_1)a + (y_2 - y_1)b = u_2 - u_1 \\ (x_3 - x_1)a + (y_3 - y_1)b = u_3 - u_1 \\ (x_2 - x_1)c + (y_2 - y_1)d = v_2 - v_1 \\ (x_3 - x_1)c + (y_3 - y_1)d = v_3 - v_1\end{cases} \tag{1}$$

Where $a, b, c, d$ could be obtained as:

$$\begin{cases}a = \dfrac{\begin{vmatrix}u_2-u_1 & y_2-y_1 \\ u_3-u_1 & y_3-y_1\end{vmatrix}}{2A(0)} \\ b = \dfrac{\begin{vmatrix}x_2-x_1 & u_2-u_1 \\ x_3-x_1 & u_3-u_1\end{vmatrix}}{2A(0)} \\ c = \dfrac{\begin{vmatrix}v_2-v_1 & y_2-y_1 \\ v_3-v_1 & y_3-y_1\end{vmatrix}}{2A(0)} \\ d = \dfrac{\begin{vmatrix}x_2-x_1 & v_2-v_1 \\ x_3-x_1 & v_3-v_1\end{vmatrix}}{2A(0)}\end{cases} \tag{2}$$

And $A(0)$ is the area of the triangle enclosed by the three sites and is retrieved as:

$$A(0) = \frac{1}{2}\begin{vmatrix}1 & x_1 & y_1 \\ 1 & x_2 & y_2 \\ 1 & x_3 & y_3\end{vmatrix} = \frac{1}{2}\begin{vmatrix}x_2 - x_1 & y_2 - y_1 \\ x_3 - x_1 & y_3 - y_1\end{vmatrix} \tag{3}$$

The horizontal divergence $D$ and vorticity $\xi$ are estimated with following equations:

$$D = \frac{(v_2-v_1)(y_3-y_1)-(u_3-u_1)(y_2-y_1)+(x_2-x_1)(v_3-v_1)-(x_3-x_1)(v_2-v_1)}{(x_2-x_1)(y_3-y_1)-(x_3-x_1)(y_2-y_1)} \tag{4}$$

$$\xi = \frac{(u_2-u_1)(y_3-y_1)-(v_3-v_1)(y_2-y_1)+(x_2-x_1)(u_3-u_1)-(x_3-x_1)(u_2-u_1)}{(x_2-x_1)(y_3-y_1)-(x_3-x_1)(y_2-y_1)} \tag{5}$$

Building upon these methods, the fine structures of mesoscale vortexes are effectively captured during various weather conditions including monsoon, convection initiation (CI), rainfall, and the evolution of convective storms (Liu et al., 2017; Guo et al., 2023).

The methods mentioned above could be applied when there are three or more CDL/RWP sites. However, only wind field observations at two stations are available during the passage of Typhoon Doksuri. We have mentioned it as a part of outlook at the end of the manuscript, as following "And it is promising to realize the convergence and divergence characteristic analysis within the ABL based on the multi-instrument network observations, which is of great importance to recognize the convection initiation and development of storms."

Reference:

[1] Liu, M. & Yang, Y. (2017). Calculation of Horizontal Divergence and Vorticity Using Wind Profiler Network. Advances in Meteorological Science and Technology, 7(1), 27-32.

[2] Guo, X., Guo, J., Zhang, D. L., & Yun, Y. (2023). Vertical divergence profiles as detected by two wind-profiler mesonets over East China: Implications for nowcasting convective storms. Quarterly Journal of the Royal Meteorological Society, 149(754), 1629-1649.

5. Is it possible to give a further discussion for conclusion (4) in the manuscript regarding on the findings on wind speeds of "15 m/s to 25 m/s on 28 July. maximum wind speed values of 51.25 m/s and 52.34 m/s." An explanation of the such findings' significance would be more interesting for the readerships instead of list the information.

AR: Thanks for your suggestion. We have added the following explanations in the section 4.4 and section 5.

section 4.4: "Before 20:00 LST on 27 July, the CDL-1 was not under the influence of the 34-kts wind circles and wind speeds within the ABL were basically in the range of 15 m/s ~ 20 m/s. Between 20:00 LST on 27 July and 03:00 LST on 28 July, obvious

differences in wind speeds existed at different heights affected by the typhoon. High wind speeds appeared around 1000 m while low wind speeds occurred below 500 m which was consistent with the distribution characteristics of the wind speed in the typhoon boundary layer."

section 5: "Before being affected by the 34-kts wind circles, there was little difference in wind speeds at different heights. Affected by the typhoon, high wind speeds appeared around 1000 m while low wind speeds occurred below 500 m which was consistent with the distribution characteristics of the wind speed in the typhoon boundary layer."

**Specific comments:**

1. Line 106, change "labeled" to "labelled".

AR: Revised. Thanks for your suggestion.

2. Line 196, it would be better that "980" and "hPa" list in the same line.

AR: Thanks for your suggestion. This sentence has been revised to "the pressure dropped to the minimum of 980 hPa approximately."

3. Line 220, remove "of two instruments".

AR: Revised. Thanks for your suggestion.

4. Line 240, change "wind shear layers appeared at 600 m during 11:00 LST ~ 13:00 LST and 1100 m during 12:00 LST ~ 13:00 LST" to "wind shear layers appeared at 600 m and 1100 m during 11:00 LST ~ 13:00 LST and 12:00 LST ~ 13:00 LST, respectively".

AR: Revised. Thanks for your suggestion.

5. Figure 6: please explain why not to set the intercept to be zero in the linear regression model since two exact same physical quantities are compared? What is the physical significance of these two intercepts individually?

AR: Thanks for your question and suggestion. In the manuscript, we choose the general linear fitting and don't set the intercept to be zero specifically. Although the CDL and the RWP both detect wind field based on Doppler effect, they have some differences in spatial-temporal resolution and instruments noise. If setting the intercept to be zero, it

may result in a large difference between the fitted curve and the trend in actual data changes. When using the general linear fitting, the actual distribution of the data and possible systematic biases between two instruments would be reflected better. The value of the two intercepts may be related to the possible fixed deviation of the two instruments.

To fully exhibit the differences in wind speed and direction measurements of the CDL and the RWP, we add the results of setting the intercept to be zero in the linear regression model in the Figure 3 as shown below. This figure has also been added in the revised paper (Figure 6).

[Figure]

Figure 3. Statistical results of the wind speed (a) and wind direction (b) from 27 July to 29 July 2024.

---

## Author Comment (AC3)

**Response to community (Chong Wang):**

**General comments:**

The tropical cyclone is the most severe meteorological disasters in southeastern of china, the background in the introduction is described in detail. This work is co-operated with China Meteorological Administration (CMA), combining with other observation methods, no-blind zone wind speed profiles is achieved.

Here are some suggestions

1. Cloud information is important during the tropical cyclone, so, the CNRs are necessarily in the figures.

AR: Thanks for your suggestion. In the revised manuscript, the variation of CDL SNR has been added in Figure 10 and Figure 11. As the CNRs of the RWP are not provided, Figure 12 has not been revised.

2. More evolution of wind field during the tropical cyclone should be discussed.

AR: Thanks for your suggestion. We have added the following explanations in the section 4.4 and section 5.

section 4.4: "Before 20:00 LST on 27 July, the CDL-1 was not under the influence of the 34-kts wind circles and wind speeds within the ABL were basically in the range of 15 m/s ~ 20 m/s. Between 20:00 LST on 27 July and 03:00 LST on 28 July, obvious differences in wind speeds existed at different heights affected by the typhoon. High wind speeds appeared around 1000 m while low wind speeds occurred below 500 m which was consistent with the distribution characteristics of the wind speed in the typhoon boundary layer."

section 5: "Before being affected by the 34-kts wind circles, there was little difference in wind speeds at different heights. Affected by the typhoon, high wind speeds appeared around 1000 m while low wind speeds occurred below 500 m which was consistent with the distribution characteristics of the wind speed in the typhoon boundary layer."